# AN EXPLORATION OF CONDITIONING METHODS IN GRAPH NEURAL NETWORKS

## ABSTRACT

The flexibility and effectiveness of message passing based graph neural networks (GNNs) induced considerable advances in deep learning on graph-structured data. In such approaches, GNNs recursively update node representations based on their neighbors and they gain expressivity through the use of node and edge attribute vectors. E.g., in computational tasks such as physics and chemistry usage of edge attributes such as relative position or distance proved to be essential. In this work, we address not what kind of attributes to use, but *how to condition* on this information to improve model performance. We consider three types of conditioning; weak, strong, and pure, which respectively relate to concatenation-based conditioning, gating, and transformations that are causally dependent on the attributes. This categorization provides a unifying viewpoint on different classes of GNNs, from separable convolutions to various forms of message passing networks. We provide an empirical study on the effect of conditioning methods in several tasks in computational chemistry.

## 1 INTRODUCTION

Graph neural networks (GNNs) are a family of neural networks that can learn from graph-structured data. Starting with the success of GCN (Kipf & Welling, 2016) in achieving state-of-the-art performance on semi-supervised classification, several variants of GNNs have been developed for this task, including Graph-SAGE (Hamilton et al., 2017), GAT (Veličković et al., 2017), GATv2 (Brody et al., 2021), EGNN (Satorras et al., 2021) to name a few most recent ones.

Most of the models based on the message-passing framework utilize conditional linear layers. We define *"conditioning"* as using additional information together with feature vectors from its neighbors' nodes. For example, EGNN (Satorras et al., 2021) conditions message vectors on the distance between two nodes or DimeNet (Gasteiger et al., 2020b) additionally utilizes angle information. Many neural network models use conditioning in their layers without exploring their different variants. Therefore, improving upon the type of conditioning could still improve most state-of-the-art models. We believe that this is the first work that analyzes different conditioning methods in GNNs.

In this paper, we categorize three conditioning methods: weak, strong, and pure. They differ in the level of dependency on the a given quantity, such as an edge attributes, and differ in complexity. Message passing neural network (MPNN) using *weak conditioning* method *concatenates* attributes with node features. In this scenario, linear layers effectively gain an attribute-dependent bias, which we consider a weak type of conditioning as this does not guarantee that the attribute is actually utilized, i.e., it could be ignored. On the other hand, we have *pure conditioning* method which forces the model to always use the attributes by letting them causally parametrize transformation matrices. However, from a practical perspective pure conditioning is computationally expensive and it can be simplified to a *strong conditioning* method, which corresponds to an attribute-dependent *gating* of the outputs of linear layers. We experiment with these three conditioning methods in variations of the EGNN model (Satorras et al., 2021) on computational chemistry datasets QM9 (Ramakrishnan et al., 2014) and MD17 (Chmiela et al., 2017) and show the advantage of strong conditioning over weak conditioning in performance, and over pure conditioning in training time.

The main contributions of this paper are:

(i) A unifying analysis of geometric message passing by formulating conditional transformations in terms of various forms of conditional linear layers.

(ii) An intuitive exposition of different conditioning methods in the context of convolutional message passing.

(iii) Empirical studies that show the benefit of *strong conditioning* methods, as well as the benefit of *deep conditioning* in multi-layer perceptron-based message functions.

In this work, we address not what kind of attributes to use, but *how to condition* on this information to improve model performance. As such, we focus on an intuitive analysis and in the experimental section we do not intend to achieve the best performance, but focus on ablation studies in order to obtain general take-home messages.

## 2 PRELIMINARIES

In this section, we introduce the relevant materials on graph neural networks on top of which we will later complement our analysis and definitions of conditioning methods.

### 2.1 GRAPH NEURAL NETWORK

In this work, we consider the graph regression task as an example. A graph is represented by $\mathcal{G} = (\mathcal{V}, \mathcal{E})$ with nodes $v_i \in \mathcal{V}$ and edges $e_{ij} \in \mathcal{E}$. A typical message passing layer (Gilmer et al., 2017) is defined as:

$$\boldsymbol{m}_{ij} = \phi_e(\boldsymbol{h}_i^l, \boldsymbol{h}_j^l, \boldsymbol{a}_{ij}) \tag{1}$$

$$\boldsymbol{m}_i = \sum_{j \in N(i)} \boldsymbol{m}_{ij} \tag{2}$$

$$\boldsymbol{h}_i^{l+1} = \phi_h(\boldsymbol{h}_i^l, \boldsymbol{m}_i) \tag{3}$$

Where $\boldsymbol{h}_i^l$ is the embedding of node $v_i$ at layer $l$, $\boldsymbol{a}_{ij}$ is the edge attribute of nodes $v_i$ and $v_j$, and $N(i)$ is the set of neighbors of the node $v_i$. Finally, $\phi_e$ and $\phi_h$ are the message (edge) and update (node) functions respectively which are commonly parametrized by Multilayer Perceptrons (MLPs).

### 2.2 GEOMETRIC GRAPH NEURAL NETWORKS

When the graphs have an embedding in Euclidean space, i.e., each node $v_i$ has an associated position $\mathbf{x}_i \in \mathbb{R}^n$, we want to leverage this geometric information whilst preserving stability/invariance to rigid-body transformations. That is, many tasks are invariant to Euclidean distance preserving transformations in $E(n)$. E.g., the prediction of energy of a system of atoms in invariant to its global position and orientation in space. Several works have shown how to build equivariant message passing based graph neural networks for such geometric graphs.

Central in those works is the conditioning of the message and update function on invariant geometric attributes, such as the pairwise distance $\mathbf{a}_{ij} = \|\mathbf{x}_j - \mathbf{x}_i\|$, as popularized in (Satorras et al., 2021), or covariant spherical/circular harmonic embeddings of relative position $\mathbf{a}_{ij} = Y(\mathbf{x}_j - \mathbf{x}_i)$ as is common steerable group convolution-based graph NNs (Brandstetter et al., 2021). Here we consider attributes that transforms predictably via representations of $E(n)$ as *covariants*, and those that remain invariant as *invariants*. Such covariants typically contain more (directional) information, but require specialized operations such as the Clebsch-Gordan tensor product (Thomas et al., 2018; Anderson et al., 2019) in order to preserve equivariance of the graph NNs. Satorras et al. (2021) show that with a simple recipe based on invariant attributes, one can often obtain equally powerful graph NNs. As such, we focus this paper on the use of $\mathbf{a}_{ij} = \|\mathbf{x}_j - \mathbf{x}_i\|$ as a sufficiently expressive attribute, and model the message and update functions $\phi_e$ and $\phi_h$ as regular MLPs.

Our objective then is to understand what is the most effective way of utilizing attributes in geometric graph NNs. To make this notion of conditioning explicit, we will denote the message and update function as

$$\phi_e(\mathbf{h}_i^l, \mathbf{h}_j^l \,|\, \mathbf{a}_{ij}) \quad \text{and} \quad \phi_h(\mathbf{h}_i^l \,|\, \mathbf{a}_i)\,,$$

where we note that, although uncommon, it is possible to define invariant or covariant geometric node attributes $\mathbf{a}_i$ (Brandstetter et al., 2021).

## 3    ANALYSIS OF CONDITIONING METHODS

In the subsequent, we unify several conditioning methods used in literature through the notion of conditional linear layers, and by discussing them in relation to the prevalent convolution layer and its variations. As a starting point, we use the fact that convolution is a simple form of message passing with linear message functions conditioned on relative position, i.e.,

$$\mathbf{m}_{ij} = \phi_e(\mathbf{f}_j^l \,|\, \mathbf{x}_j - \mathbf{x}_i) = \mathbf{W}(\mathbf{x}_j - \mathbf{x}_i)\mathbf{f}_j \,, \tag{4}$$

and the following update function typically is the application of a point-wise activation function $\sigma$, i.e., $\phi_h(\mathbf{f}_i) = \sigma(\mathbf{f}_i)$, possibly with a skip connection as in ResNets He et al. (2016). In general, geometric graph NN do not just *linearly* transform node features, but generally do this *non-linearly* via message/update functions parametrized by MLPs, leading to a notion of non-linear convolutions when the attributes are invariant/covariant quantities (Brandstetter et al., 2021).

Importantly, these MLPs themselves are parametrized by linear layers, intertwined with non-linear activation functions, and nothing prevents from conditioning each of these linear layers on the attributes. It is the purpose of this paper to categorize several options for conditioning and understand what effect this has on performance.

### 3.1    CONDITIONAL LINEAR LAYERS

We propose the following modifications of the linear layer, as to make them conditional on attributes

$$\mathbf{W_a h} := \mathbf{W\,h} \qquad\qquad\qquad\qquad no \tag{5}$$
$$\mathbf{W_a h} := \mathbf{W\,(h \oplus a)} \qquad\qquad\qquad weak \tag{6}$$
$$\mathbf{W_a\,h} := (\mathbf{W}^a \mathbf{a}) \odot (\mathbf{W}^h\,\mathbf{h}) \qquad\qquad strong \tag{7}$$
$$\mathbf{W_a h} := \mathbf{W(a)h} \qquad\qquad\qquad\qquad pure \tag{8}$$

where to keep the similarity to the common notation for linear transformations, we use the notation $\mathbf{W_a}$ to denote that the linear transformation is conditioned on $\mathbf{a}$. We further use to notation $\mathbf{h} \oplus \mathbf{a}$ to denote the concatenation of vectors $\mathbf{h}$ and $\mathbf{a}$, and use $\odot$ to denote element-wise multiplication, i.e., $\mathbf{a} \odot \mathbf{b} = (a_1 b_1, a_2 b_2, \dots)^T$.

We stress the hierarchy in terms of the dependence of the transformation on $\mathbf{a}$. The most direct dependence is in the *pure* method, in which the transformation is causally parametrized by $\mathbf{a}$, followed by *strong* conditioning in which a standard unconditional transformation $\mathbf{W}^h\,\mathbf{h}$ is *gated* by a vector $\mathbf{W}^a\,\mathbf{a}$. Both the pure and strong methods are by construction forced to utilize the attribute, as, in particular in the strong case, the transformation would not exists without attribute $\mathbf{a}$. We refer to equation 6 as *weak* conditioning, as in principle the transformation would still exist in the absence of the attribute, and moreover, if the dimensionality of $\mathbf{h}$ is much larger than that of $\mathbf{a}$, the transformation of the attribute only contributes to a small extent to the output of this layer. We hypothesize that this hierarchy correlates with performance and experimentally test this hypothesis in Sec. 5. What follows is a brief analysis of these types of conditioning.

### 3.2    PURE CONDITIONING CORRESPONDS TO BI-LINEAR LAYERS

As common for implementations of convolutions, one typically expands the convolution kernel $\mathbf{W}(\mathbf{x}_j - \mathbf{x}_i) \in \mathbb{R}^{d_o \times d_i}$, i.e., a transformation matrix with elements $W_{oi}(\mathbf{x}_j - \mathbf{x}_i)$ that depend on relative position, in a basis $\{\phi_b : \mathbb{R}^n \to \mathbb{R}\}_b^{d_b}$ via

$$W_{oi}(\mathbf{x}_j - \mathbf{x}_i) = \sum_b^{d_b} W_{boi}\, \phi_b(\mathbf{x}_j - \mathbf{x}_i)\,. \tag{9}$$

The basis could be the usual $3 \times 3$ pixel basis, or it could be a continuous basis for when the continuous structure of the data is to be respected, such as circular or spherical harmonics (Worrall et al., 2017; Weiler & Cesa, 2019; Thomas et al., 2018; Anderson et al., 2019), B-splines (Bekkers, 2020; Fey et al., 2018), or hermite polynomials (Sosnovik et al., 2020). In recent works on the parametrization of continuous functions as Neural Fields (Xie et al., 2022) or in transformer-based

methods (Vaswani et al., 2017), such basis functions are often referred to as *coordinate embeddings* or *position encodings*.

An important observation is that, given such a parametrization through basis functions, the pure conditioning layer corresponds to a bi-linear layer

$$h_o^{l+1} = \sum_b \sum_i \phi_b(\mathbf{a}) \, W_{boi} \, h_i^l \quad \Longleftrightarrow \quad \mathbf{h}^{l+1} = \phi(\mathbf{a}) \overset{bilinear}{\mathbf{W}} \mathbf{h}^l, \tag{10}$$

where we use $i$ to index the input feature vector, $o$ the output feature vector, and $b$ the basis functions.

Such bilinear layers are often implicitly used in convolutional architectures, where the expansion in the basis function is typically hard-coded or pre-computed. On continuous data, such as point cloud methods, the bilinear layer is ubiquitous. Notably, in the context of steerable group equivariant convolutions, the transformations happen via bilinear operators called the Clebsch-Gordan tensor product, in combination with spherical harmonic embeddings of relative position (Brandstetter et al., 2021). Outside of the (group) convolution literature, bilinear layers are often used to explicitly model conditioning on geometric attributes, for which a relevant example to our current work is DimeNet (Gasteiger et al., 2020b). DimeNet uses an advanced message passing framework in which messages and updates are conditioned on geometric quantities (embedded as spherical harmonics and radial basis functions) via combinations of weak, strong, and pure conditioning.

### 3.3 STRONG CONDITIONING CORRESPONDS TO DEPTH-WISE SEPARABLE CONVOLUTIONS

In later works, DimeNet was improved by DimeNet++ (Gasteiger et al., 2020a), both in performance and speed, by replacing the compute-heavy bilinear layers (pure conditioning) with the more efficient gating type conditioning (strong conditioning). The computational bottleneck of pure conditioning motives the use of strong conditioning, a route that has proven successful in convolutional architectures computer vision as well, via the use of so-called depth-wise separable convolutions (Sifre & Mallat, 2014; Chollet, 2017).

Chollet (2017) shows huge efficiency gains, both in terms of computing and performance, when factorizing convolution kernels in two parts. One part does the channel mixing, which does not depend on the relative position, while another part depends on the relative position which scales/gates the output. That is if the kernel is given by

$$W_{oi}(\mathbf{x}_j - \mathbf{x}_i) = W_o^a(\mathbf{x}_j - \mathbf{x}_i) W_{oi}^h, \tag{11}$$

the convolution boils down to message passing with conditional linear layers of the *strong* type, as we can write

$$\mathbf{h}^l = (W^a(\mathbf{x}_j - \mathbf{x}_i)) \odot (\mathbf{W}^h \mathbf{h}), \tag{12}$$

where we can define $\mathbf{W}^a(\mathbf{x}_j - \mathbf{x}_i) = \mathbf{W}^a \mathbf{a}_{ij}$ as the linear transformation of a coordinate embedding $\mathbf{a}_{ij} = \phi(\mathbf{x}_j - \mathbf{x}_i)$ if we want the connection to equation 7 explicit. The fact that the transformation overall is linear and that it splits into a part that does and a part that doesn't depend on pair-wise attributes allows for very efficient implementations that first perform a group-wise convolution, followed by channel mixing. This principle is at the core of the recently popularized ConvNeXt architecture (Liu et al., 2022) and plays an essential role in equivariant graph NNs on (molecular) point clouds in order to be able to scale up (Thomas et al., 2018). Separability recently also proved necessary in order for equivariant convolutional NNs to scale up to large groups, such as the scale-rotation-translation group (Knigge et al., 2022). In the context of conditional NN to parametrize *neural fields* (Xie et al., 2022), strong conditioning commonly appear in the form of so-called *film layers* Perez et al. (2018).

### 3.4 WEAK CONDITIONING CORRESPONDS TO LINEAR LAYERS WITH A CONDITIONAL BIAS

Weak conditioning (equation 6) corresponds to a standard linear layer with an adaptive bias:

$$\mathbf{W_a} \mathbf{h} = \mathbf{W}(\mathbf{h} \oplus \mathbf{a}) = \underbrace{\mathbf{W}' \mathbf{h}}_{\text{no condition}} + \underbrace{\mathbf{W}'' \mathbf{a}}_{\text{conditional bias}} = \mathbf{W}' \mathbf{h} + \mathbf{b}(\mathbf{a}), \tag{13}$$

where $\mathbf{W}'$ are the first $d_h$ rows of $\mathbf{W}$ that are applied to the $\mathbf{h} \in \mathbb{R}^{d_h}$ of the concatenated vector, and $\mathbf{W}''$ are the last $d_a$ rows of $\mathbf{W}$ that are applied to $\mathbf{a} \in \mathbb{R}^{d_a}$. This simple form of conditioning

is most used in literature to condition any type of NN on the conditioning vector $\mathbf{a}$, from message passing methods (Gilmer et al., 2017; Satorras et al., 2021), to conditional neural fields, e.g. as a simple but effective form of modulation in sirens (Dupont et al., 2022), to conditional variational auto-encoders (Sohn et al., 2015).

## 3.5 CONDITIONAL MLPs

With the various forms of conditional linear layers given in equations 6, 7, and 8 we can build conditional MLPs, by simply replacing the usual linear layers with the conditional ones. Such MLPs could e.g. be denoted with $\mathrm{MLP}(\mathbf{h} \,|\, \mathbf{a})$. Usually, only the first layer of such a conditional MLP is conditional and usually of the weak type, as with EGNN (Satorras et al., 2021). In the experiments we show that this simple choice is sub-optimal in the context of geometric message passing à la EGNN, and show that improvements can be made by either by conditioning more layers, or switching to strong conditioning.

## 4 RELATED WORK: GEOMETRIC MESSAGE PASSING FOR COMPUTATIONAL CHEMISTRY

In the previous section, we discussed several options for conditioning message/update functions for us in message passing graph neural networks, as well as their use in other fields of deep learning. In our experiments we benchmark the three conditioning methods (equations 6, 7, 8) in the context geometric message passing for computational chemistry. Recent works in this category, and the types of conditioning used in those works, are as follows.

EGNN (Satorras et al., 2021) is a message passing neural network (MPNN) that uses a *weak* conditioning method to utilize the distance between nodes. DimeNet (Gasteiger et al., 2020b) is the type of MPNN, where message embeddings interact based on the distance between atoms and the angle directions. *Pure* conditioning is adapted to utilize angle directions in message update and aggregation. Tensor Field Network (Thomas et al., 2018) is a neural network for 3D point clouds. Each point in TFN is associated with a vector in a representation of $SO(3)$. To condition one representation on another, a tensor product of representations is used which in its general form corresponds to *pure* conditioning. However, many works of the steerable message passing kind, including TFN and NequIP (Batzner et al., 2022), implement a *separable* variation (*strong* conditioning) for the sake of computational efficiency. The exception is steerable EGNN (Brandstetter et al., 2021), which uses steerable MLPs with *pure* conditioning in each layer, i.e., not only the first layer of the message function as in EGNN.

DimeNet++ (Klicpera et al., 2020) is an extension of DimeNet which changed conditioning method from pure to strong to increase efficiency. Klicpera et al. (2020) showed that changing the conditioning method decreased the runtime time of the original DimeNet by a factor of 5. SchNet (Schütt et al., 2017) is another example of MPNN that utilizes atom locations using a strong conditioning method. PaiNN (Schütt et al., 2021) further extends SchNet by projecting the interatomic distances via radial basis functions and iteratively updating the vectors along with the scalar features but also using a strong conditioning method.

## 5 EXPERIMENTS

In this section, we design experiments to evaluate the effectiveness of three conditioning methods shown in equation 6, 7, and 8 on two real-world datasets: QM9 (Ramakrishnan et al., 2014) and MD17 (Chmiela et al., 2017). We want to demonstrate the effect of the choice of the conditioning method and present the results of some other models on benchmarks to provide context. In our experiments, we report Mean Absolute Error (MAE) between model predictions and ground truth.

**Implementation details** As a baseline architecture, we use the EGNN model that consists of 7 layers, 128 features per hidden layer, and a 2-layer message (edge) function $\phi_e = \mathrm{MLP}(\mathbf{h}_i \oplus \mathbf{h}_j \,|\, \|\mathbf{x}_j - \mathbf{x}_i\|^2)$ with only weak conditioning via equation 6 in the first layer. We will test modifications of this model by changing the weak conditioning, to strong or pure, and explore the effect of conditioning multiple layers in the edge function MLP.

We found it useful to pass geometric distances through two-layer MLP, as a form of vectorization/coordinate embedding of the pairwise distance, for the QM9 dataset and Random Fourier Feature layer (Rahimi & Recht, 2007) for the MD17 dataset.

We use the same values for hyperparameters from the EGNN paper (Satorras et al., 2021) for both datasets: we trained each model on QM9/MD17 dataset for a total of 1.000/500 epochs, used Adam optimizer, batch size 96 (64 for pure-EGNN), weight decay 1e-16, and cosine decay for the learning rate starting at 5e-4.

**QM9**  The QM9 (Ramakrishnan et al., 2014) dataset consists of small molecules represented as a set of atoms (up to 29 atoms per molecule), each atom having a 3D position associated and a five-dimensional one-hot node embedding that describe the atom type (H, C, N, O, F). The QM9 dataset has 3D coordinate locations of each atom and we use the distance between two atoms (nodes) as an edge attribute. The dataset comprises 12 quantum properties for each of the molecules. We use 100k molecules for training, 18k for validation, and 13k for testing.

Table 1 shows the Mean Absolute Error for the prediction of 12 molecular properties for weak and strong conditioning methods. We can clearly see that using *strong conditioning shows lower MAE than weak conditioning for all properties by an average of 10%*. Due to a large number of data points and our limited computational resources we limit our experiments to weak and strong conditioning methods. Training one epoch of pure-EGNN was on average 267x longer than training strong-EGNN, which made it intractable to validate pure conditioning.

Table 1: Mean Absolute Error for the molecular property prediction benchmark in QM9 dataset.

| Task
Units | $\alpha$
$bohr^3$ | $\Delta\epsilon$
meV | $\epsilon_{HOMO}$
meV | $\epsilon_{LUMO}$
meV | $\mu$
D | $C_v$
cal/mol K | $G$
meV | $H$
meV | $R^2$
$bohr^3$ | $U$
meV | $U_0$
meV | $ZPVE$
meV |
|---|---|---|---|---|---|---|---|---|---|---|---|---|
| SchNet | .235 | 69 | 43 | 38 | .030 | .040 | 19 | 17 | .180 | 20 | 20 | 1.50 |
| DimeNet++ | .044 | 33 | 25 | 20 | .030 | .023 | 8 | 7 | .331 | 6 | 6 | 1.21 |
| EGNN (baseline) | .071 | 48 | 29 | 25 | .029 | .031 | 12 | 12 | .106 | 12 | 11 | 1.55 |
| weak-EGNN (ours) | .067 | 50.52 | 28.11 | 25.74 | .032 | .031 | 9.81 | 10.51 | .138 | 11.14 | 9.94 | 1.455 |
| strong-EGNN (ours) | .061 | 44.52 | 27.48 | 23.83 | .023 | .029 | 9.79 | 10.29 | .088 | 9.39 | 9.88 | 1.45 |

The advantage in the performance of strong conditioning over weak conditioning is clearly shown in Table 1. We also hypothesize that the advantage of the strong conditioning method will be more evident in constrained settings of shallow networks. In order to test whether strong conditioning provide a more efficient parametrization then weak conditioning, we repeat experiments from Table 1 on three molecular properties, but with a more shallow network. Table 2a shows the MAE for the prediction of three molecular properties for weak and strong conditioning methods for models with 7 and 3 layers. On average improvement of strong conditioning over weak conditioning was more highlighted for the shallow network than for the deep network (10.8% vs 7.7%). It demonstrates the importance of conditioning methods in models with less capacity.

We define conditioning depth as the number of conditional layers in the message function. In our experiments, the conditioning depth was 2 as we conditioned all two layers in the message function. We test the effect of conditioning depth for different conditioning methods. Table 2b shows the MAE for the prediction of five molecular properties for weak and strong conditioning methods for models with conditioning depths 1 and 2. The weak-EGNN shows better performance with single conditional layer while strong-EGNN benefits more from two conditional layers.

**MD17**  MD17 (Chmiela et al., 2017) is a dataset of eight small organic molecules containing up to 17 total atoms composed of the atoms H, C, N, O, F. For each molecule, an ab-initio molecular dynamics simulation was run using DFT to calculate the ground state energy and forces. At intermittent timesteps, the energy, forces, and configuration (positions of each atom) were recorded. We uniformly sample 50k molecules for training, 10k for validation, and 10k for testing.

Table 3 shows the Mean Absolute Error for the prediction of energies for 8 molecules for three conditioning methods. From Table 3 we can see that strong conditioning improved the performance of 5 molecules by an average of 14.5%. It did not show improvement in molecules that initially had high MAE. In experiments with pure conditioning, to decrease computational cost we used half of the model: $2 \rightarrow 1$ conditional layers, hidden embedding size of $128 \rightarrow 64$, and $7 \rightarrow 3$ layers. With

Table 2: Mean Absolute Error (MAE) for the molecular property prediction benchmark in QM9 dataset for different numbers of layers and conditioning depths. $\Delta$ is the percentage difference of MAE, lower $\Delta$ is better.

(a) shallow vs. deep models

| Task
Units | $\alpha$
$bohr^3$ | $\Delta\epsilon$
meV | $\epsilon_{HOMO}$
meV |
|---|---|---|---|
| weak-EGNN (7 layers) | .067 | 50.52 | 28.11 |
| strong-EGNN (7 layers) | .061 | 44.52 | 27.48 |
| $\Delta$ | -9.22% | -11.88% | -2.24 % |
| weak-EGNN (3 layers) | .076 | 61.26 | 36.48 |
| strong-EGNN (3 layers) | .069 | 53.25 | 32.9 |
| $\Delta$ | -9.68% | -13.08% | -9.81% |

(b) 1 vs. 2 conditional layers

| Task
Units | $\alpha$
$bohr^3$ | $\Delta\epsilon$
meV | $\epsilon_{HOMO}$
meV | $\epsilon_{LUMO}$
meV | $\mu$
D |
|---|---|---|---|---|---|
| weak-EGNN (cond. depth=1) | .061 | 46.48 | 28.59 | 22.76 | .024 |
| weak-EGNN (cond. depth=2) | .067 | 50.52 | 28.11 | 25.74 | .032 |
| strong-EGNN (cond. depth=1) | .066 | 45.72 | 27.16 | 22.41 | .026 |
| strong-EGNN (cond. depth=2) | .061 | 44.52 | 27.48 | 23.83 | .023 |

these changes, a pure-EGNN model was 14.3x and 16x times slower than strong-EGNN and weak-EGNN respectively. The negative impact of decreasing the number of layers on the performance of strong-EGNN on the QM9 dataset can be seen in Table 2a. Considering that we reduced by half the number of layers, hidden embedding size, and conditioning depth, pure-EGNN shows competitive performance with weak/strong-EGNN on 5 molecules and outperforms it on some of them. This potential gain in performance might, however, not outweigh the computation costs of pure over strong or weak conditioning.

Table 3: Mean Absolute Error (MAE) for the conformational energies (meV) prediction benchmark on MD17 dataset.

| Molecule | Aspirin | Benzene | Ethanol | Malonaldehyde | Naphthalene | Salicylic acid | Toluene | Uracil |
|---|---|---|---|---|---|---|---|---|
| Cormorant | 4.25 | 0.997 | 1.171 | 1.778 | 1.258 | 2.862 | 1.474 | 0.997 |
| sGDML | 8.239 | 4.336 | 3.035 | 4.336 | 5.204 | 5.204 | 4.336 | 4.77 |
| SchNet | 5.204 | 3.035 | 2.168 | 3.469 | 4.77 | 4.336 | 3.903 | 4.336 |
| weak-EGNN | 25.608 | 4.262 | 2.576 | 11.755 | 13.693 | 19.132 | 16.447 | 12.468 |
| strong-EGNN | 29.168 | 3.265 | 2.278 | 10.454 | 13.794 | 15.131 | 18.804 | 11.726 |
| pure-EGNN* | 34.706 | 3.513 | 4.582 | 12.403 | 21.421 | 27.221 | 14.692 | 12.249 |

* pure-EGNN network is smaller than weak/strong-EGNN.

## 6 CONCLUSION

In this work, we explore how graph neural networks can recursively update node embeddings with edge attribute information such as geometric distance. We provide a unifying analysis of several works in literature that utilize attributes, through a notion of conditional linear layers. We present three conditioning methods to this end: weak, strong, and pure. Weak conditioning method concatenates edge attributes to node features, strong conditioning method gates node features, and in pure conditioning, edge attributes causally parametrize transformation matrices. We explain the intuition of each method and apply them to the EGNN model to empirically show their difference in performance and computational cost on QM9 and MD17 datasets.

Our conclusion is that strong conditioning (gating) generally beats weak conditioning (concatenation) in the message passing framework. We also conclude that pure conditioning is computationally prohibitive in geometric message passing, whilst it can achieve competitive performance with weak and strong conditioning methods with a smaller network. This confirms the impact observed by other works on separable convolutions, s.a. in Depth-wise separable convolutions (Chollet, 2017) and ConvNeXt (Liu et al., 2022), and justifies the use of separable (group) convolutions of steerable tensor-based methods such as TFN (Thomas et al., 2018). While these methods can be formulated as linear message passing methods of the convolutional form, we show that performance gains can be achieved through multi-layer conditional message functions, as a form of non-linear convolution (Brandstetter et al., 2021). In this setting, we show that it can be beneficial to condition all layers in the conditional MLP, rather than only condition the first layer as is the convention.

We believe that our categorization of conditioning methods, combined with our empirical findings can be used as a guideline in designing the next generation of geometric graph neural network architectures.

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
