# OpenReview forum: "An Exploration of Conditioning Methods in Graph Neural Networks"
_ICLR.cc/2023/Conference — Submitted to ICLR 2023_

### Official Review · Reviewer_QMHG · 2022-10-23

**Confidence:** 3
**Clarity, Quality, Novelty And Reproducibility:** 1. Notation is confused, for example,…
**Correctness:** 3
**Technical Novelty And Significance:** 2
**Empirical Novelty And Significance:** 2
**Recommendation:** 3

**Strength And Weaknesses:**

Strengths:

1. A general framework of conditioning methods is proposed.


Weaknesses:

1. In the experimental section, it would be great if the authors could show more statistics on the results, like standard deviation, so that it can further show that improvement made by strong conditioning is statistically significant. Also, some experimental setups are missing, like the number of runs for the two tasks.
2. Even though the authors empirically justify that for the EGNN instance, conditioning methods help boost performance, can any theory be developed to verify it?
3. Since the authors only show the advantage of strong conditioning over weak conditioning on the EGNN model, it is not convincing enough to assert that this will still hold for other models. Therefore, It is encouraged and interesting to see the applications of these conditioning methods on a few more models.


Questions:

1. Are these conditioning methods designed only for geometric GNNs, i.e., GNNs with graph data input with Euclidean embedding?
2. In Table 2(b), is there any reason why strong-EGNN is much worse than weak-EGNN in predicting $\mu$?
3. Is there any reason why EGNN is not compared on the MD17 dataset?





**Summary Of The Paper:**

The paper proposes three types of conditioning (weak, strong, and pure) to unify some conditioning methods in the literature. Two empirical testing studies show that strong conditioning can perform better than weak conditioning and more efficiently than pure conditioning.

**Summary Of The Review:**

Overall, it is insightful to unify current models into a framework. However, more numeric and theoretical supports are encouraged.

---

### Official Review · Reviewer_xVZv · 2022-10-24

**Confidence:** 3
**Clarity, Quality, Novelty And Reproducibility:** they are OK.
**Correctness:** 3
**Technical Novelty And Significance:** 1
**Empirical Novelty And Significance:** 1
**Recommendation:** 3

**Strength And Weaknesses:**

In this paper, the authors analyze the three message-passing schemes of Graph Neural Networks conditional on the geometric property, namely weak, strong, and pure methods, which relate to concatenation, gating, and transformations depending on the geometric attributes. To compare the effects of these methods, the authors perform experiments on molecular datasets and conclude that the strong method, i.e., the gating method, not only has higher computational efficiency but also achieves better performance in most cases.

Strength:
1. The paper is well-written and easy to read.
2. The paper focuses on an interesting problem, i.e., how to employ geometric attributes in GNNs properly.

Weaknesses:
1. The novelty and significance of this paper are concerned. As mentioned in the paper, the three methods have already been raised in previous work. And this paper only summarizes these methods and only gets some trivial results through experiments without theoretical analysis. The paper proposes an interesting question, but it doesn't lead to a meaningful conclusion.
2. The experiment is not sufficient. The authors only compare the three message-passing methods based on the EGNN model. In order to get a more reasonable conclusion, it is necessary to conduct experiments on several basic models.


**Summary Of The Paper:**

In this paper, the authors analyze the three different message-passing schemes of Graph Neural Networks conditional on the geometric property, namely weak, strong, and pure methods, and empirically study their effect.

**Summary Of The Review:**

This paper studies an interesting problem in geometric GNN design. However, it has limited novelty and has drawn few meaningful conclusions.

---

### Official Review · Reviewer_j1mL · 2022-10-25

**Confidence:** 3
**Correctness:** 3
**Technical Novelty And Significance:** 2
**Empirical Novelty And Significance:** 2
**Recommendation:** 3

**Clarity, Quality, Novelty And Reproducibility:**

The clarity of the paper is good. However, the quality and novelty of the proposed method are limited. The code is not provided for reproducibility.

**Strength And Weaknesses:**

Pros:
1. The existing ways of using additional information when updating node embeddings in GNNs are summarized and analyzed.
2. The paper is easy to follow and well-organized.

Cons:
1. The addressed methods are all based on existing works without having new conditioning methods.
2. The experiments are only done based on EGNN, which is not a well strong model in the related field. Besides, the empirically compared conditioning methods are only focusing on geometric distances.
3. The models except for EGNN and EGNN variants in Table 1 and Table 3 seem to be unrelated to the discussions. These models perform much better than the EGNN variants and the authors do not mention them at all.

**Summary Of The Paper:**

In this paper, the authors summarize and analyze the existing ways (or called conditioning methods) that use edge attribute information together with node embeddings in GNN. The intuition of each conditioning method is explained for comparison. In experiments, the authors use a GNN model as the baseline to show the difference between these conditioning methods on benchmark datasets.

**Summary Of The Review:**

Even though the existing conditioning methods in GNNs are summarized and analyzed, the authors only provide limited experiments. It would be better to use more powerful GNN as baseline and more comprehensively evaluate the conditioning methods.

---

### Decision · Program_Chairs · 2023-01-20

**Decision:**

Reject

**Justification For Why Not Higher Score:**

limited technical contribution

**Justification For Why Not Lower Score:**

n/a

**Metareview: Summary, Strengths And Weaknesses:**

This paper considers three types of conditioning to improve the performance of GNNs and apply the proposed method to computational chemistry tasks. While the method is intuitive and easy to understand, the technical contribution of conditioning is limited, as these conditioning methods exist in the literature. The experiments need to be expanded to incorporate models other than EGNN and the significance of the results needs to be provided in the results table. Given these concerns, I would recommend rejection of this paper.